# The Italian External Quality Assessment Program for Cystic Fibrosis Sweat Chloride Test: Does Active Participation Improve the Quality?

**DOI:** 10.3390/ijerph17093196

**Published:** 2020-05-04

**Authors:** Marco Salvatore, Annalisa Amato, Giovanna Floridia, Federica Censi, Gianluca Ferrari, Fabrizio Tosto, Rita Padoan, Valeria Raia, Natalia Cirilli, Giuseppe Castaldo, Ettore Capoluongo, Ubaldo Caruso, Carlo Corbetta, Domenica Taruscio

**Affiliations:** 1Undiagnosed Rare Diseases Interdepartmental Unit, Istituto Superiore di Sanità, National Center Rare Diseases, 00161 Rome, Italy; amatoann@libero.it (A.A.); fcensi@iss.it (F.C.); g.ferrari@areasrl.com (G.F.); fabrizio.tosto@iss.it (F.T.); domenica.taruscio@iss.it (D.T.); 2Unità di Bioetica, Istituto Superiore di Sanità, 00161 Roma, Italia; giovanna.floridia@iss.it; 3Centro di Supporto Fibrosi Cistica, Dipartimento Pediatrico, Università di Brescia, ASST Spedali Civili, 25123 Brescia, Italia; ritaf54@gmail.com; 4Centro Regionale Fibrosi Cistica, Unità di Pediatria, Dipartimento di Scienze Mediche Traslazionali, Università degli Studi di Napoli, Federico II, 80145 Napoli, Italia; raia@unina.it; 5Centro di Riferimento Fibrosi Cistica, Dipartimento Materno Infantile, AOU Ospedali Riuniti, 60121 Ancona, Italia; natalia.cirilli@ospedaliriuniti.marche.it; 6Dipartimento di Medicina Molecolare e Biotecnologie Mediche, Università di Napoli Federico II; CEINGE-Biotecnologie avanzate, scarl, 80131 Napoli, Italia; giuseppe.castaldo@unina.it (G.C.); edotto70@gmail.com (E.C.); 7LABSIEM—Laboratorio per lo Studio degli Errori Congeniti del Metabolismo, Università degli Studi di Genova—DINOGMI, 16132 Genova, Italia; ubaldo.caruso48@gmail.com; 8SC Laboratorio di Riferimento Regionale per lo Screening Neonatale—Regione Lombardia, ASST Fatebenefratelli Sacco, 20157 Milano, Italia; cfm.corbetta@tiscali.it

**Keywords:** cystic fibrosis, external quality assessment, quality, sweat test, diagnosis, choloride, active participation

## Abstract

(1) Background: Diagnostic testing for cystic fibrosis (CF) is based on a sweat chloride test (SCT) considering the appropriate signs and symptoms of the disease and results of a gene mutation analysis. In 2014, the Istituto Superiore di Sanità (ISS) established a pilot Italian external quality assessment program for CF SCT (Italian EQA-SCT), which is now a third party service carried out by the ISS. (2) Methods: The ongoing scheme is prospective, enrollment is voluntary, and the payment of a fee is required. Results are shared through a dedicated web-facility. Assessment covers the analysis, interpretation, and reporting of results. (3) Results: Thirteen, fifteen, sixteen, and fifteen different laboratories, respectively, participated from 2015 to 2016 and from 2018 to 2019 in the Italian EQA-SCT scheme. Eleven different laboratories participated each year in all four rounds of the Italian EQA-SCT. (4) Conclusions: The overall results obtained from the laboratories participating constantly clearly show that their qualitative and quantitative performance improved significantly. This is due to the opportunity—after receiving the EQA results—to constantly review their performance and address any inconsistencies. We firmly believe that participation in the EQA program will improve the quality of participating laboratories and that EQA participation should become mandatory as a fundamental requirement for laboratory accreditation.

## 1. Introduction

Cystic fibrosis (CF, OMIM 219700) is the most common life-threatening autosomal-recessive disease affecting Caucasians in the western world. The highest prevalence is found in the western populations of Europe, North America, and Australia, with the disease occurring among approximately 1 in 2500 to 3500 live births or even less (e.g., Scandinavia has estimated CF incidence of 1 in 4900 live births). Treatment advances have led to improved patient outcomes. A child born with CF in the United Kingdom (UK) may live up to more than 50 years of age, but the median age of death remains 31 [1,2,3]. The median age at death in Italy is 37.2 years, according to 2016 data from the Italian Registry for Cystic Fibrosis [4].

More than 2000 genetic variants have been identified in the gene that codes for the protein cystic fibrosis transmembrane conductance regulator (CFTR); however, only 352 out of 432 variants included in CFTR2 are disease causing (https://cftr2.org/mutations_history).

The diagnostic testing of CF is based on a sweat chloride test (SCT), considering the appropriate signs and symptoms, and the results of newborn screening and gene alteration analyses. An SCT consists of a quantitative measurement of electrolytes (typically chloride) in sweat, and its indications include a phenotype suggestive of CF, a family history of CF, a positive newborn screening test, and suspicion of an atypical phenotype [5,6,7]. 

In the majority of CF patients with typical clinical features and carriers of pathogenic variants (PVs), a sweat test by quantitative pilocarpine iontophoresis test will show elevated values of sweat chloride. 

The sweat test is considered to be the gold standard for the diagnosis of cystic fibrosis [8]. Bensliman et al. [9] confirm that a sweat test should be performed on patients with pulmonary manifestations. 

In other conditions that do not fulfil the diagnostic criteria for CF, a sweat test can give an intermediate result. In these situations, both a sweat chloride test and nasal potential difference (NPD) versus intestinal current measurement (ICM) testing may be recommended [8,10,11]. The diagnosis of CF can remain uncertain in patients with suggestive clinical features but only an intermediate sweat test and no identified PVs. Rarely, a sweat test returns normal results in a patient with two identified CF-causing mutations [12,13,14] since the CFTR protein’s main function consists of transporting chloride ions in and out of the cell. In a sweat gland, defective CFTR proteins result in the impaired reabsorption of chloride ions and high levels of chloride in sweat [15]. 

The approach to this test is essential for a correct CF diagnostic pathway [16]. Italian recommendations for the execution and interpretation of sweat tests indicate the appropriate way to analyze and interpret the test results [17]. Nevertheless, due to the current lack of an Italian Accreditation Program for Sweat Test Laboratories (adherence to which is still voluntary not mandatory), the exact number of Italian laboratories performing sweat tests nationwide is not known. Therefore, it is extremely important to monitor the activity of laboratories performing SCTs in order to assure the level of accuracy of their results.

The number of laboratories performing SCT in Italy is estimated to be 37. This number includes laboratories within the Italian Referral Care Centers for Cystic Fibrosis (established by Italian Law 548/93) and some newborn screening laboratories. In addition, there are laboratories not in close collaboration with the reference Centers for the care of CF in various Italian regions that perform sweat tests.

In this respect, the areas of inconsistency in current practices for SCT highlight the need to provide specific national initiatives aimed at improving the general performance of sweat tests [18,19]. In order to increase and monitor quality in laboratories performing SCTs, an Italian External Quality Assessment program for SCTs (EQA-SCT) was established in 2014 at the Italian National Center for Rare Diseases (CNMR) of the Istituto Superiore di Sanità (ISS, Rome) [20,21]. In 2015, this program was recognized as institutional and, therefore, permanent [22]. 

This program was designed to cover the analysis of SCTs in Italian clinical laboratories measuring the concentration of chloride in sweat, which is used for the diagnosis of CF.

EQA programs are valuable management tools to monitor and improve laboratory quality by assessing a laboratory’s use of diagnostic technologies in clinical routines. The aim of such programs is mainly educational, focused on continuous quality improvement, particularly for accredited laboratories. Accuracy in the analysis, interpretation, and reporting of results is fundamental for a correct diagnosis. Thus, staff operators should be well trained in this area. EQA provides the correct instruments to educate and ameliorate all phases involved in an efficient diagnostic service, thus improving the internal quality of a laboratory [23,24].

The aim of this study is to focus on the results of eleven different laboratories belonging to the Italian Referral Care Centers for Cystic Fibrosis network, which participated continuously in four rounds of the Italian EQA-SCT (the 2015 to 2016 and 2018 to 2019 schemes).

## 2. Materials and Methods 

The Italian EQA-SCT program was fully described by Salvatore et al. [20,21]. The program started as a pilot project in 2014; in 2015, it was recognized as institutional and became permanent with participation opened to all Italian laboratories. Figure 1 shows a flow chart of the Italian EQA-SCT scheme’s activities over a year. The scheme organizer (ISS) periodically assesses the registered participating laboratory. Laboratories can access the scheme all the year, but registration for each annual scheme occurs from April to July due to logistical reasons. Three different samples are prepared, dispatched, and analyzed (one sample per month on a fixed date) by participating laboratories from August to late December of each year, and the evaluation of results is performed from February to March the following year.

Overall, thirteen, fifteen, sixteen, and fifteen different laboratories, among the 37 Italian CF public Referral Care Centers, participated, respectively, in the Italian EQA-SCT schemes from 2015 to 2016 and from 2018 to 2019 (Figure 2). One or more participating laboratories from different CF public Referral Care Centers can be present in each Italian region. An Identification Number (ID) is assigned to each laboratory by the scheme organizer. All data from the laboratories are managed through a dedicated web-facility [21,22].

In the present paper, the laboratories participating in the Italian EQA-SCT are identified by a progressive alphabetical ID from a to s. 

### 2.1. Assessment

The Italian EQA-SCT assessment team involves experts in CF diagnosis who assess laboratory performance against their definition of a satisfactory performance. The panel of experts is composed of representatives from both ISS and the National Scientific Societies involved in cystic fibrosis research, clinics, and management (SIFC, Italian Cystic Fibrosis Society; SIMMESN, Italian Society for the Study of Inborn Metabolic Diseases and Newborn Screening; SIBiOC, Italian Society of Clinical Biochemistry and Clinical Molecular Biology). The results are first assessed online (by using a dedicated web-facility) and then re-assessed collegially in dedicated meetings. All data are treated in an anonymous way. Assessment covers analytical performance, quantitative analytical performance, and qualitative descriptions of the structures. Results are evaluated according to the Italian guideline indications [17]. 

The panel of assessors decides in advance the marking criteria for the key elements/points expected to be in the submitted results and assigns a quantitative score for each key point. 

To analyze performance, technical adequacy in performing a sweat chloride test (including the stimulation method, sweat collection, and analytical method) is evaluated. Scores range from 0,0 to 3,3 for each of the three parameters [20]. 

The quantitative analytical performance analysis includes an evaluation of the reporting information, chloride concentration values, and clinical sensitivity [20]. The evaluation of reports is based on the presence of the following information: identification data, date of test, date of sample collection, date of report, weight and volume of sweat collected, an indication of insufficient collection (< 75 mg), stimulation method, analyte(s), analytical method, chloride reference intervals (normal results if < 40 mEq/L; < 30 mEq/l in subjects less than 6 months of age; intermediate results if 40−60 mEq/L, 30−60 mEq/L in subjects less than 6 months of age; abnormal if > 60 mEq/L), interpretation of results, presence of report signature; and clear legibility of the clinical report. Scores range from 0,0 to 10,0 for the evaluation of each report (maximum total score = 30,0) [20].

Chloride concentration values exceeding 75% of the reference values are not included in the analyses. A 20% of error is accepted for an expected value of 20 mEq/L; a 10% error is accepted for an expected value of 100 mEq/L. A proportional percentage of errors is considered for all values between 20 and 100 mEq/L. Scores range from 0,0 to 10,0 for the evaluation of each report (maximum total score = 30,0) [20].

The clinical sensitivity assessment evaluates the consistency of the sweat chloride results with a correct range and clinical interpretations of the results. Score ranges from 0,0 to 10,0 for the evaluation of each report (maximum total score = 30,0) [20].

Finally, a portion of the assessment is dedicated to a qualitative description of the diagnostic laboratories. This report involves the evaluation of an ad hoc on-line pre-test questionnaire administered to all participants and including (i) the number of tests performed each year in the laboratory, (ii) the number of tests performed each year by a dedicated technician, and (iii) the time of test execution. No score ranges are assigned for this assessment, but a comment is eventually sent to the laboratory [20]. 

The maximum total score is fixed at 100. The omission of key points results in a demerit from the total score for the EQA. The same error is scored once and, if necessary, that score is associated with a dedicated comment. 

In 2016, poor performance criteria were established: poor performance is assigned to laboratories that (i) exceed more than 50% of the reference values in chloride concentration titration, (ii) obtain incorrect titration values due to an unintentional sample exchange and/or clerical or transcription errors, (iii) submit a report where the interpretation is missing or wrong, and/or (iv) submit reports where fundamental information is missing. 

### 2.2. Samples

For each scheme, three sweat-like-samples (SLS) were commercially prepared (LTA s.r.l., Milano, Italy) and dispatched to participating laboratories. The samples consisted of an aqueous material mimicking normal sweat composed of dipotassium carbonate, lactic acid, carbamide, glucose, albumin, thimerosal, demineralized water, and NaCl (the latter at established range concentrations). 

Two independent working units quantified and validated all samples before the dispatch. The samples were labeled with specific identification codes (e.g., “Sample I-EQA-SLS-1”, “Sample I-EQA-SLS-2”, and “Sample I-EQA-SLS-3”) and dispatched with mock clinical information and technical data [20,21]. The laboratories were required to analyze the samples according to their routine procedures. Before analysis, the samples were kept at 4–8 °C to avoid evaporation and treated according to relevant guidelines (e.g., when sweat is collected onto filter paper, the elution time should be greater than 1 min and less than 3 hours, samples should be homogenized and mixed thoroughly before analysis, etc.) [17].

## 3. Results

A total of 12 different samples (3 for each scheme) were dispatched between 2015 and 2019: one sample within the normal chloride reference range, four samples within the pathologic chloride reference range, one sample with a non-physiological quantity of chloride (>150 mmol/L), and one sample within the pathologic chloride reference range but whose sweat quantity (mg) was insufficient (< 75 mg) to perform the test (Table 1) [17].

Overall, thirteen, fifteen, sixteen, and fifteen different laboratories among the 37 Italian CF public Referral Care Centers participated, respectively, from 2015 to 2016 and from 2018 to 2019 in the Italian EQA-SCT (Figure 2). Eleven laboratories (a, c, e, f, g, h, i, j, k, l, and m) participated in all the Italian EQA-SCT rounds from 2015 to 2019. Three laboratories participated in one EQA-SCT round (b, d, and s); three laboratories participated for two rounds (p, q, and r); and two laboratories participated for three rounds (namely n and o) (Figure 3).

Figure 4 shows the total scores of the laboratories not participating in all Italian EQA-SCT rounds (a) and of those participating constantly from 2015 to 2019 (b). The average scores of all laboratories increased constantly year by year from 69,3/100,0 in the 2015–2016 scheme to 93,8/100,0 in the 2018–2019 scheme (Figure 5). In particular, the technical adequacy assessment scores progressively increased over the four years. The evaluation included the stimulation method, sweat collection, and analytical method. For the stimulation method, the laboratories preferred to use gel discs containing pilocarpine nitrate at 2–5 g/L at both electrodes. 

During the test, sweat was collected onto pre-weighed chloride free filter paper or gauze. Although colorimetry, coulometry, and ISEs are satisfactory methods for the analysis of sweat chloride, the Italian EQA-SCT participating laboratories preferred to use colorimetry or coulometry.

An important improvement is evident in the average chloride concentration score (from 12,0/30,0 in the 2015–2016 scheme to 27,3/30,0 in the 2018–2019 scheme) (Figure 6). In particular, in the 2015–2016 scheme, seven out of eleven different laboratories (a, f, g, i, j, k, and m) made 12 critical errors in their chloride titration (score range per sample = 0,0/10,0 to 0,9/10,0). In the 2016–2017 scheme, the accuracy of chloride titration improved dramatically. The average score was 22,7/30,0, and only three errors were made by three different laboratories (f, g, and k), with the score range per sample increasing from 0,0/10,0 to 0,6/10,0. Notably, more than 57% of laboratories that made only one mistake in the 2015–2016 scheme in chloride titration did not show that same mistake in the following years, and the remaining 43% made only one error per laboratory. 

In the 2017–2018 EQA round, only one error was made (laboratory k, score 0,0/10,0). The score average was 25,8/30,0, and the score per sample was high for almost all participants, ranging from 2,3 to 10,0/10. In the 2018–2019 scheme, an average score of 27,3/30,0 was achieved, and none of the laboratories obtained a critical score for each sample.

The clinical sensitivity average scores, evaluated by comparing the consistency of the sweat chloride result with the correct range and clinical interpretation of the result, were heterogeneous during all periods. Nevertheless, the overall results increased from the 2015–2016 to the 2018–2019 EQA schemes (Figure 6). 

The average clinical sensitivity score in the 2015–2016 EQA was 20,0/30,0. In particular, one single laboratory (laboratory k) offered the incorrect interpretation of sample I-EQA-SLS-1 (inappropriate use of the adopted reference intervals) and I-EQA-SLS-2 (inappropriate use of adopted reference intervals). Eight laboratories out of eleven (c, e, g, h, j, k, l, m) returned a negative score (0,0/10,0) for sample I-EQA-SLS-3’s interpretation (the laboratories did not report that the chloride value was not physiological and, consequently, that the result should have been questioned) [20]. 

In the 2016–2017 EQA scheme, clinical sensitivity’s average score was higher than that in the 2015–2016 scheme (26,4/30,0). One single laboratory (laboratory k) was marked with a “0” score for all three samples since their interpretation of the results was not present in the report [20]. In the 2017–2018 EQA, the average clinical sensitivity score was 24,1/30,0. Two different laboratories were marked with 0,0/30,0 in the I-EQA-SLS-1 evaluation (laboratory c and laboratory m; incorrect classification of results considering the clinical information provided) and one laboratory in I-EQA-SLS-2 (laboratory k; the absence of a request to repeat the test on a borderline sample). In the 2018–2019 EQA, the average clinical sensitivity score was 27,3/30,0. One laboratory (laboratory k) reported an incorrect interpretation of the results in both samples I-EQA-SLS-2 and I-EQA-SLS-3, and a 0,0/10,0 score was consequently assigned. 

The sweat test reporting average score ranged from 27,6/30,0 in 2015 to 29,3/30,0 in the 2018−2019 scheme (Figure 6). Here, “interpretation of results”, “reference intervals”, “weight and volume of sweat collected”, “date of primary sample collection”, “analytical method”, and “interpretation of results” were the most frequently missing information [20].

Poor performance was assigned to laboratory k in both the 2015–2016 and 2016–2017 EQA-SCT rounds due to its lack of interpretation of the results in all samples (SLS-1, SLS-2, and SLS3). Furthermore, this laboratory obtained a “0” score in its chloride concentration titration for sample SLS-3 [20]. In the 2017–2018 scheme, no poor performance was assigned to the laboratories participating in the Italian EQA-SCT. In the 2018–2019 round, poor performance was assigned to laboratory k due to its incorrect interpretation of the results in samples I-EQA-SLS-1 and 2.

## 4. Discussion

Sweat testing involves the quantitative analysis of sweat chloride available to establish a CF diagnosis. In the context of the appropriate signs and symptoms of the disease, this test remains the gold standard for the diagnosis of CF, even in the genomics area [25,26].

Today, a positive newborn screening for CF, clinical signs suggestive of CF, or a family history of CF [27,28,29] are routinely accepted as the main indicators for sweat testing.

A sweat chloride concentration test is the analysis of choice, even with respect to the genotype; it is particularly important in case of a confirmed CF diagnosis in a patient presenting a positive CF screen, including diagnosis (CFSPID)/CFTR related metabolic syndrome (CRMS) or CFTR related disorders. For this reason, the urgent need to standardize the collection and analysis of sweat has clearly been expressed by several papers in the last few years [20,21].

Recent papers showed areas of inconsistency in current sweat testing practices in Italy and Europe, highlighting the need for evidence based national guidelines to improve the practice and its management strategies, thus requiring that laboratories standardize the execution, interpretation, and reporting of the relevant results [18,19,30,31,32,33].

EQA programs are fundamental to improving the efficiency and the quality of laboratory services. The EQA assesses if a laboratory is compliant with nationally and internationally established criteria, standards, and guidelines. During the EQA assessment, errors may occur. These errors may be due to incorrect information in the reporting results (e.g., name and age of the patient, incorrect reporting of correct results of analysis, etc.) [23,24]. In the case of an SCT, for example, it is extremely important to correctly report the age of a patient since reporting different reference intervals may result in completely different interpretations, thereby resulting in a lower mark (poor performance) [17]. The identification of all possible errors is, therefore, extremely important to minimize the possibility of systematically repeating those errors to improve the final quality of the results [23,24]. 

For this purpose, to assess the qualitative descriptions of the laboratories participating in the Italian EQA-SCT every year, an ad hoc on-line pre-test questionnaire is administered to all participants [20], including information on the number of tests performed in the previous year, the number of tests performed in the previous year by a dedicated technician, and the timeframe from sweat collection to sweat analysis. 

Technical quality assessment ensures that a systematic approach to technical preparation and execution is adopted. In particular, the adequacy of performing a sweat chloride test (including stimulation method, sweat collection, and analytical method) is evaluated by a panel of experts. Almost all laboratories participating in all four years of the Italian EQA-SCT scheme attained the maximum score during the evaluation, confirming that proper indications by dedicated guidelines are routinely followed by the laboratories that perform sweat test analyses [14,17].

EQA performance is primarily based on the ability of a laboratory to assay the expected results with respect to defined and appropriate limits [23,24]. In this regard, an important improvement was achieved in the average chloride concentration score among the laboratories that constantly participated in the Italian EQA-SCT. Indeed, the average score increased 2,3-fold from 2015–2016 the scheme (median score = 12,0/30,0) to the 2018–2019 scheme (median score = 27,3/30,0). 

The Italian EQA-SCT clinical sensitivity assessment is based on an evaluation of the consistency of a sweat chloride result with the correct range and a clinical interpretation of the results. An interpretation of the results should be consistent with indications of the diagnostics and/or the resulting significance of the results [23,24]. 

The most frequent errors in the 2015–2019 period were due to (a) an inappropriate use of the adopted reference intervals; (b) a lack of the indication of the impossibility to reach a diagnostic interpretation of the results for a non-physiological chloride value; (c) the complete absence of an interpretation of the results in the report; (d) an incorrect classification of the chloride results relative to the provided clinical information; and (e) a complete absence of a request to repeat the test when the chloride value was in a borderline range. 

Nevertheless, the average score increased by 1.4-fold from 2015 to 2018 in the laboratories that actively and constantly participated in all Italian schemes.

The reporting assessment highlighted the general heterogeneity in the modality of result reporting. The most frequently missing information concerned “reference intervals”, “date of sample collection”, and (in particular) “interpretation,” which affected clinical sensitivity. A complete report should recommend and explain a “normal”, “borderline”, and “positive” result and the reason why the referral of the patient for additional counselling is appropriate [17,34]. 

## 5. Conclusions

The overall results obtained from the eleven laboratories with ongoing participation in the Italian EQA-SCT clearly show that the laboratories’ qualitative and quantitative performance improved significantly, thereby also improving their clinical testing/reporting. This result is likely due to the fact that after having received the EQA results, the laboratory has the opportunity to review its performance and address any inconsistencies. If performance is identified, the EQA providers should identify the corrective actions needed to improve the quality of the diagnostic service. 

Although it is not possible to ensure that errors due to methodological, equipment, or technical problems [34] will not occur in the coming years, the data suggest that the laboratories participating in our schemes experienced significant improvement in performance, thus encouraging long-term participation. In particular, the Italian EQA-SCT organizers strongly encouraged poorly-performing laboratories to review their internal processes and contact the working groups dedicated to sweat tests within the Italian Cystic Fibrosis Society. Two annual face-to-face meetings (one by the ISS and one by the Italian Cystic Fibrosis Society) were also organized to discuss the EQA-SCT results with the laboratories. Single occurrences of poor performance should be considered isolated incidents and used as an opportunity to review all internal procedures. A careful evaluation of each error may determine whether there is a system failure that could be used as a new occasion to re-design a test or to calibrate instruments and adjust training procedures. 

It will be interesting to monitor the results in future studies [20,21]. We firmly believe that compliance with national and international established criteria, as well as participation in the EQA program, will improve the quality of participating laboratories and that EQA participation should become a mandatory and fundamental requirement of laboratory accreditation.

## Figures and Tables

**Figure 1 ijerph-17-03196-f001:**
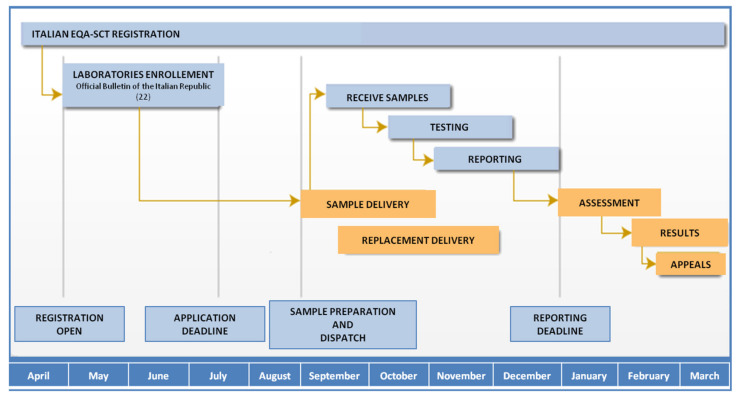
Italian EQA-SCT scheme calendar.

**Figure 2 ijerph-17-03196-f002:**
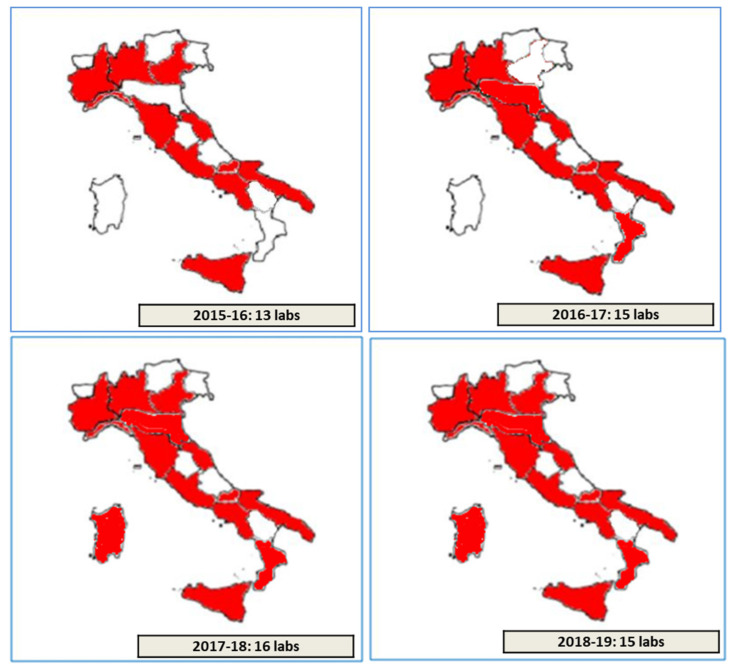
Number and regional distribution of laboratories participating within the period 2015–2019.

**Figure 3 ijerph-17-03196-f003:**
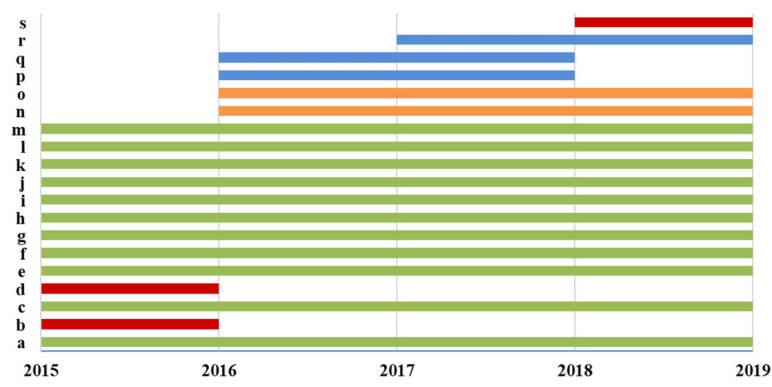
Participation in the Italian EQA-STC.

**Figure 4 ijerph-17-03196-f004:**
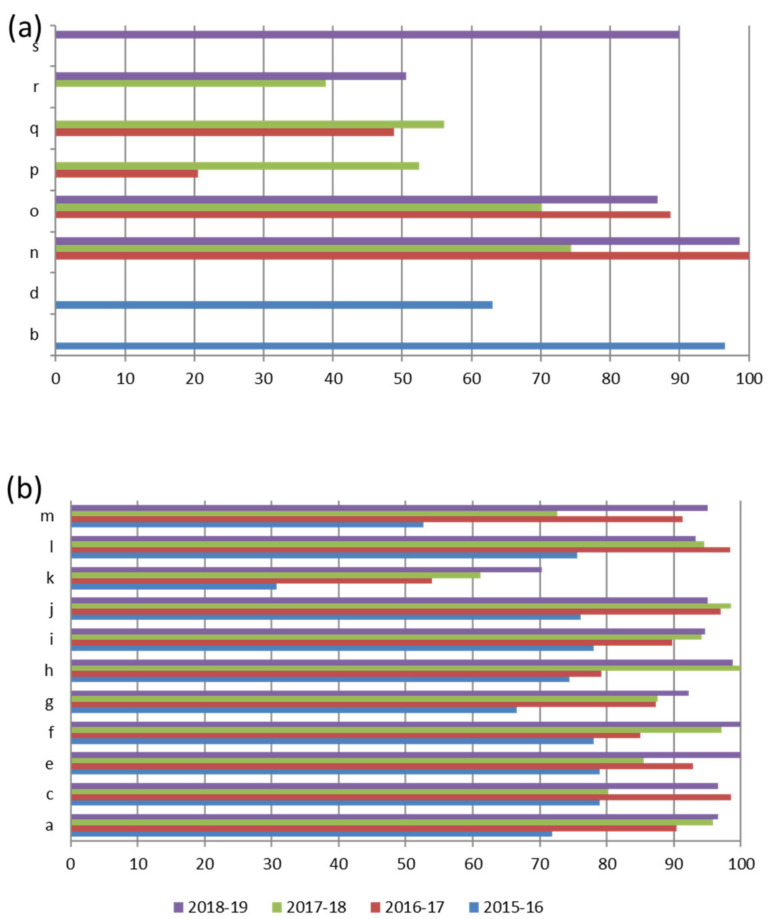
Total score of Laboratories not participating in all Italian EQA-SCT rounds (**a**) and of those participating in the 2015–2016 and 2018–2019 rounds (**b**).

**Figure 5 ijerph-17-03196-f005:**
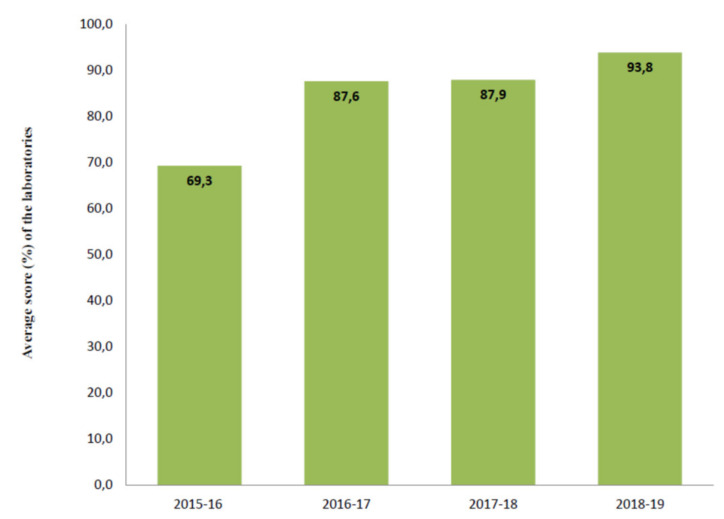
Average score of the eleven laboratories participating in all Italian EQA-SCT rounds.

**Figure 6 ijerph-17-03196-f006:**
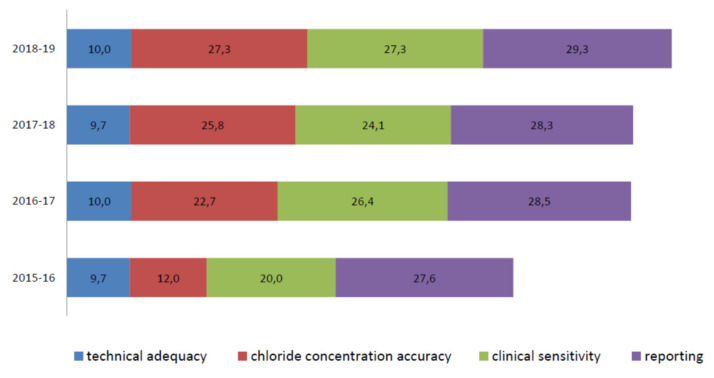
Average score details of the eleven laboratories participating in the Italian EQA-SCT from 2015 to 2019.

**Table 1 ijerph-17-03196-t001:** Characteristics of the 12 different samples dispatched between 2015 and 2019.

		2015−16	2016−17	2017−18	2018−19
**I-EQA-SLS-1**	Expected Chloride concentration (mEq/L)	20	35,66	31,6	48
	Mock chloride quantity (mg)	120	86	180	100
	Mock age	3 years	30 days	2 months	60 years
	Expected interpretation	normal	borderline	borderline	borderline
**I-EQA-SLS-2**	Expected Chloride concentration (mEq/L)	40	54,96	61,6	69,25
	Mock chloride quantity (mg)	80	120	80	60
	Mock age	2 months	62 years	42 years	1 month
	Expected interpretation	borderline	borderline	pathologic	pathologic BUT sweat quantity not sufficient to perform test
**I-EQA-SLS-3**	Expected Chloride concentration (mEq/L)	200	110,72	104,0	80,75
	Mock chloride quantity (mg)	130	92	280	130
	Mock age	16 years	20 years	20 days	8 years
	Expected interpretation	pathologic BUT non-physiological quantity of chloride (>150 mmol/L)	pathologic	pathologic	pathologic

Legend: the Italian EQA-SCT experts and coordinators dispatched samples with mock clinical information and technical data (namely “Expected Chloride concentration”, “Mock chloride quantity” and “Mock age”) to test the participant’s consistency of a sweat chloride result with the correct range and the clinical interpretation of the result (“Expected interpretation”).All the laboratories that participated in our survey, routinely adopt intra-laboratory quality control systems to monitor the accuracy and precision of their procedure. The results of patients (including those related to our survey) are considered valid and reported when the accuracy and precision parameters obtained from the control cards are < 2 SDs.

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
