# Peer review of "The Italian External Quality Assessment Program for Cystic Fibrosis Sweat Chloride Test: Does Active Participation Improve the Quality?"

_ijerph, 2020, doi:10.3390/ijerph17093196_

Round 1

Reviewer 1 Report

The article

“The Italian External Quality Assessment program for Cystic Fibrosis Sweat Chloride Test: does an active participation improve the quality?” by

  1. Salvatore, A. Amato, G. Floridia, F. Censi, G. Ferrari, F. Tosto, R. Padoan, V. Raia, N. Cirilli, G. Castaldo, E. Capoluongo, U. Caruso, C. Corbetta and D. Taruscio

describes an Italian quality assessment program for CF sweat tests carried out from 2015-2018 in up to 16 different laboratories.

The program is based on experience from a pilot program in 2014 carried out by the Istituto Superiore di Sanità (ISS).

The program includes:

  • Analysis
  • Interpretation
  • Reporting results

It is a prospective study and enrollment was voluntary (payment fee required).

The conclusion of the study overall is that qualitative and quantitative performance of sweat tests improved significantly.

Overall:

Sweat test is the gold standard test for cystic fibrosis. It is therefore of major importance to have a secure quality testing program. Test performance and levels are very important to have compliant with national as well as international standard criteria.

The quality assessment program has previously been described and published by Salvatore et al.

The EAQ is well described in methods and results.

Importance of continued assessment should be stressed.

Minor comments:

Maybe simple, but a few words of preparation before testing could be added – hydration and temperature of patient, etc.

Line 47: occurring among approximately 1 in 2500 to 3500 live births – or even less as fx Scandinavian countries

Line 141: Score ranges from 0 to 10 – but differs from figures in Fig. 6

Line 201: participating laboratories use colorimetry, coulometry. – either or?

Conclusions: suggest that - compliance with national and international established criteria, - is also added to the final conclusions.

Comments to Figures and tables:

Figure 2.

Number and regional distribution of laboratories participating within period 2015-19.

Are 2017-18 and 2018-19 identical? – the difference for 15 and 16 not clear.

Table 1.

Needs further explanation in legend –

Figure 3

– legend to tell what colours indicate (1 month participation, 2 months etc..)

Figure 4 – may be omitted

Figure 5.

Total score of Laboratories not participating to all Italian EQA-SCT turns (a) and of those participating from 2015-16 to 2018-19 turn (b) - there is no a and b – please explain

Figure 6.

Average score detail of eleven laboratories participating to Italian EQA-SCT from 2015 to 2019. Technical adequacy: include stimulation method, sweat collection, and analytical method. Score ranges from 0 to 3,3 per parameter. Chloride concentration, clinical sensitivity, reporting: score range from 0 to 10 per sample

– but scores of Chloride concentration, clinical sensitivity, reporting were between 12 and up to almost 30?? – please explain

Reviewer 2 Report

The manuscript describes an external quality assessment program for cystic fibrosis sweat chloride testing in Italy and the improvement of participating laboratories over time. This is an important subject matter as it is necessary for clinical laboratories to provide accurate and timely diagnostic test results. It is concerning that some laboratories perform poorly in the EQAP and yet are providing diagnostic results for CF patients.

The English writing of the manuscript needs attention throughout the manuscript.

Minor comments:

Line 50-51 – “The median age is much higher in Italy” Please state the median age.

In the Introduction section please explain why there is a lack of an Italian Accredited Program for Sweat Test Laboratories?

What was the reason for some laboratories discontinuing their participation (e.g. Labs b and d)?

It seems that 4 months is  along turn-around-time for reporting proficiency testing results. Is there a reason labs are given from September to December to report?

A limitation of the study is that although laboratories may have improved their EQAP scores, it is not clear that their clinical testing/reporting has improved.

Reviewer 3 Report

The article entitled “The Italian External Quality Assessment program for Cystic Fibrosis Sweat Chloride Test: does an active participation improve the quality?” by Salvatore etal., is an interesting study to provide a database for the  external quality assessment of sweat chloride test for Cystic Fibrosis, tested among different laboratory between 2015-16 and 2018-19. Overall, a quality manuscript suitable for publication with minor modifications.

In the results section rearrange the paragraphs, avoid short ones having just 2-3 lines.

Could the authors provide a table having comparison of expected chloride concentration, observed values with accuracy and precision?

Table 1: remove the background color and increase the font size

Figure 3: remove border line, increase the font size, add y-axis title, and bold the axes lines,

Figure 5: remove the gridlines, border, and add axes titles, seems (b) figure is missing.

Figure 6: remove the borders, indicate the titles.

Line 31: “Results: Thirteen, fifteen, sixteen, and fifteen different “ is this line correct sentence? Please check the similar sentence at line 111, and 184.

Line 187: “Although colorimetry, coulometry and ISEs” should be Although colorimetry, coulometry, and ISEs.

Author Response

“The Italian External Quality Assessment program for Cystic Fibrosis Sweat Chloride Test: does an active participation improve the quality?” by

  1. Salvatore, A. Amato, G. Floridia, F. Censi, G. Ferrari, F. Tosto, R. Padoan, V. Raia, N. Cirilli, G. Castaldo, E. Capoluongo, U. Caruso, C. Corbetta and D. Taruscio

Comments and replies to Reviewer #3

The article entitled “The Italian External Quality Assessment program for Cystic Fibrosis Sweat Chloride Test: does an active participation improve the quality?” by Salvatore etal., is an interesting study to provide a database for the  external quality assessment of sweat chloride test for Cystic Fibrosis, tested among different laboratory between 2015-16 and 2018-19. Overall, a quality manuscript suitable for publication with minor modifications.

In the results section rearrange the paragraphs, avoid short ones having just 2-3 lines.

Results section has been rearranged and the short paragraphs have been merged as suggested.

Could the authors provide a table having comparison of expected chloride concentration, observed values with accuracy and precision?

The reviewer raised a relevant point. All the laboratories that participated in our survey, routinely adopt intra-laboratory quality control systems to monitor the accuracy and precision of their procedure. The results of patients (including those related to our survey) are considered valid and reported when the accuracy and precision parameters obtained from the control cards are <two SDs. We now report this point under the Table 1. We apologize for not including the table as requested. As known Italy is still locked down and we are working from home with not a few difficulties. Unfortunately at the present time we are not able to recover all data necessary to draw the table requested.

Table 1: remove the background color and increase the font size

The table has been changed as suggested and replaced in the text.

Figure 3: remove border line, increase the font size, add y-axis title, and bold the axes lines

Figure 3 has been changed as suggested and replaced in the text.

Figure 5: remove the gridlines, border, and add axes titles, seems (b) figure is missing.

Figure 5 has been changed as suggested and replaced in the text. Title of Figure 5 is now correct; the previous one was due to a “past and copy” error. We apologies for this inconvenience.

Figure 6: remove the borders, indicate the titles.

Figure 6 has been changed as suggested; the title is now correctly indicated “Figure 6. Average score detail of eleven laboratories participating to Italian EQA-SCT from 2015 to 2019”. Additional info are now included in a “Legend to figure 6” included.

Line 31: “Results: Thirteen, fifteen, sixteen, and fifteen different” is this line correct sentence? Please check the similar sentence at line 111, and 184.

The sentence is correct and is reffered to the number of laboratories participating to I-EQA in the investigated years (2015-19) respectively.

Line 187: “Although colorimetry, coulometry and ISEs” should be Although colorimetry, coulometry, and ISEs

The sentence has been changed as suggested